# Less is more

Mengran Wu[1] and Zhiyang Liu[1]

[1] College of Electronic Information and Optical Engineering, Nankai University
liuzhiyang@nankai.edu.cn

**Abstract.** This paper focuses on developing a deep learning method for kidney and the kidney tumors and cysts. To further fuse the semantic information and the spatial information, contrast attention is added at the skip connections of a classical U-Net. To make a lighter model, we use a shallower U-Net with 4 times down strides. Evaluation results indicates that the proposed method is able to achieve a comparable performance with nnU-Net while using much fewer parameters.

**Keywords:** Kidney segmentation, deep learning, contrast attention

## 1 Introduction

A precise and quantitative evaluation on kidney masses has been an effective way in guiding future treatments. In clinical practice, such task is usually performed manually, which is time consuming and tedious. To make it reproducible, it is urgent to develop an automatic method for kidney mass segmentation and distinguish the tumors and cysts.

One of the most popular method in biomedical image segmentation is U-Net[5], which have shown great performance in almost all biomedical image segmentation tasks. The skip connections between the encoder and decoder layers enable it to gather both semantic information from the deeper layers and the spatial information from the shallower layers. The nnU-Net[1], on the other hand, utilizes the U-Net structure, but focused on the preprocessing and data augmentation methods and built an self-adaptive framework for 3D medical image segmentation tasks. The success of nnU-Net not only proved the strong representation ability of U-Net, but also highlighted the importance of proper preprocessing, especially when the images are anisotropic.

From the released training dataset of KiTS2021, we made several observations. 1) The CT images are anisotropic, and the slice spacings vary from about 0.5mm to 5mm; 2) the physical field of view varies, where some images include lung, while some only include abdomen organs; and 3) the kidney segmentation is much easier than the kidney masses. Motivated by the above observations, we propose to adopt a two-stage segmentation method as shown in Fig. 1. In particular, we extract the rough kidney region from down-sampled low-resolution images in the first stage, and generate the fine segmentation results in the second stage.

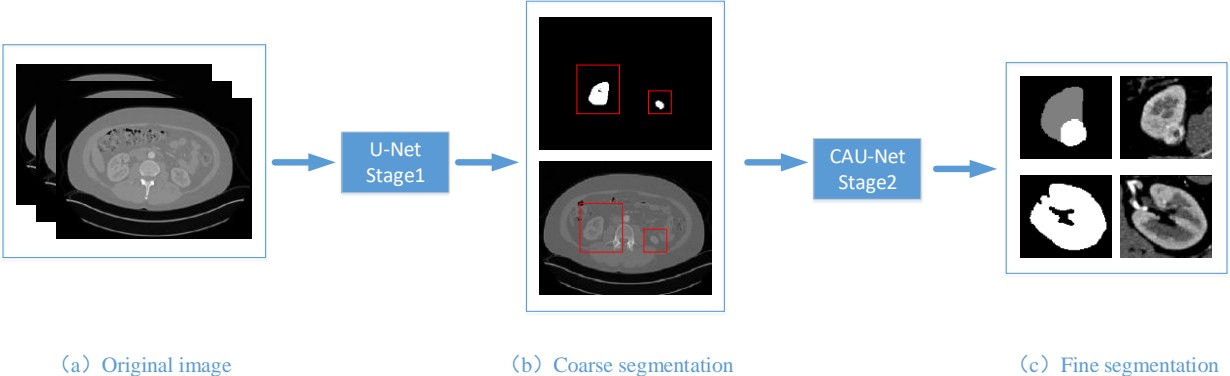

（a）Original image      （b）Coarse segmentation      （c）Fine segmentation

Fig. 1. Whole pipeline of the proposed segmentation method.

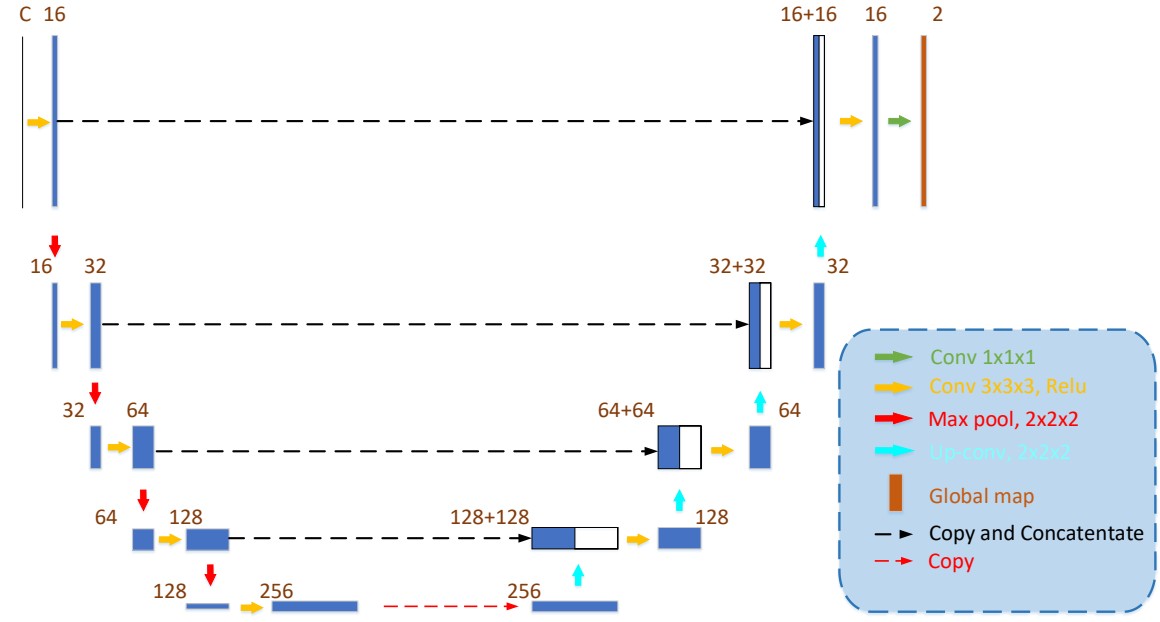

Fig. 2. 3D U-Net used for rough segmentation in this project.

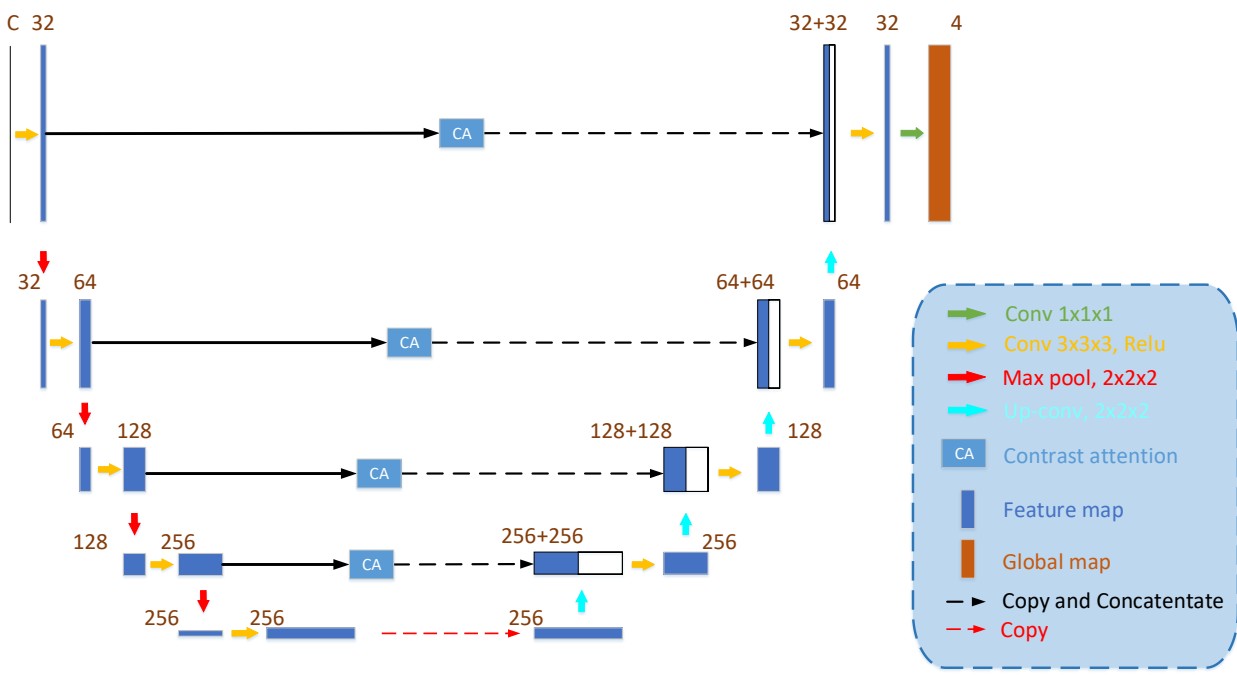

Fig. 3. Architecture of the proposed CAU-Net.

Attention mechanism has shown its great potential in both natural language processing and computer vision. In medical image segmentation tasks, several variants of U-Net have been proposed by introducing attention mechanism to U-Net[2-4]. The attention mechanism provides a guidance to the network in focusing on the most important features, and is able to increase the segmentation accuracy. Moreover, by guiding the network to extract important features, the attention mechanism is helpful in increasing the parameter efficiency. In this paper, we introduce the contrast attention mechanism to the skip connections of U-Net. As we will show in this paper, the contrast-attention U-Net (CAU-Net) is able to achieve comparable performance with nnU-Net by using, however, much fewer number of network parameters.

## 2 Methods

In this paper, we proposed a two-stage segmentation method, as shown in Fig. 1. In particular, in the first stage, a rough segmentation results for kidney were generated, which is used to extract the regions of interest (ROIs), i.e., the regions with kidney. In the second stage, the CNN focuses only on the ROIs and generates the segmentation map. In the first stage, we used 3D U-Net for rough segmentation, and in the second stage, we adopted the proposed CAU-Net for fine segmentation. The network structures of these two stages are shown in Fig. 2 and Fig. 3, respectively.

### 2.1 Training and Validation Data

Our submission made use of the official KiTS21 training set alone.

### 2.2 Preprocessing

In KiTS21, the images have varies voxel spacing. As the CNNs are not capable in interpreting the voxel spacing, we resample all images to a common voxel spacing. The choice of voxel spacing is in general a tradeoff between the textural information and spatial contextual information, due to the fact that the training of 3D CNNs are patch-based, instead of image-based. For instance, with a smaller voxel spacing, despite that a richer textural information can be preserved, the image patch of a certain matrix size would correspond to a smaller physical volume, leading to a small physical field of view in the CNNs. Therefore, the voxel spacing should be carefully chosen to arrive at a good tradeoff.

In particular, in the first stage, as the goal is coarse segmentation for locating the kidney area, we adopted a larger voxel spacing of $3.4 \times 1.7 \times 1.7$ mm. In the second stage, to generate a fine segmentation result, a voxel spacing of $0.85 \times 0.85 \times 0.85$ mm was adopted. During resampling, the masks were resampled using nearest neighbor interpolation. The images, however, adopted varies interpolations on different directions. By noting that the images in the training set are generally with small voxel spacing in the transverse plane (y-z plane), bilinear interpolation was adopted on the y-z plane. In the x-axis, as the voxel spacing varies from 0.5mm to 5mm, to reduce resampling artifacts, nearest neighbor interpolation was adopted.

From the Tab.1, the HU value for air is about -200; for bone it is typically above 400+; for kidney it is from 25 HU to 50 HU; for water it is approximately from -10 HU to 10 HU; and for blood it is from 3 HU to 14 HU. As the HU values are quantitative, we clipped the HU values to the range [-79, 304]. Then the values were subtracted by 101 and divided by 76.9. All samples in the training set were used.

In our experiment, the images were cropped to patches with size (144, 128, 128) in the first stage, and size (96,96,96) in the second stage. In the stage 1, to balance the patches with foreground and background, we ensure that at least 1/3 of the patches include foreground. In stage 2, as we have cut the kidney volumes beforehand, the patch size of (96, 96, 96) is large enough to ensure that each patch contains foreground pixels, and therefore random spatial crop is adopted.

Table 1. Typical tissues radiodensities of human body.

| Tissue | HU |
|---|---|
| Air | -200 |
| Bone | 400+ |
| Kidney | 25~50 |
| Water | 0 ±10 |
| Blood | 3~14 |

### 2.3 Proposed Method

The proposed Contrast Attention U-Net (CAU-Net) employs a U-Net like structure in general. In classical U-Net, skip connections are employed to fuse the feature maps hierarchically with the decoder feature maps. In our proposed network, contrast attention (CA) module is added at the skip connections to encourage the network extract features from higher and lower levels feature maps. As shown in Fig. 3, the proposed Contrast Attention U-Net (CAU-Net) employs a U-Net like structure in general, but makes several important modifications. The detail hyperparameters,

such as strides and kernel sizes can be find in Figs. 3-4. To make a lighter model, we use a half of filters at each convolution layers.

### 2.3.1 Contrast Attention

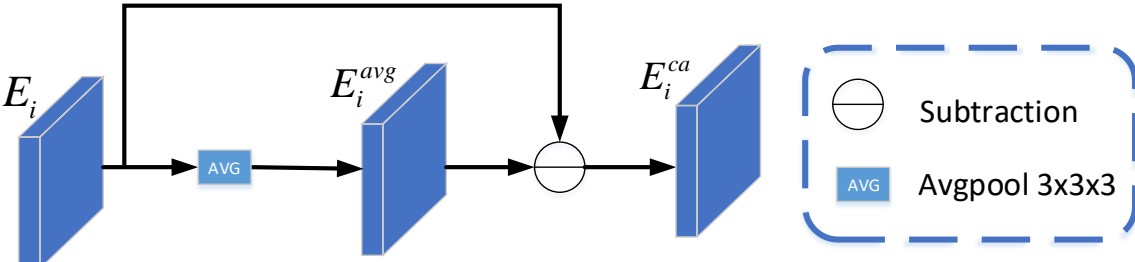

Fig. 4. The contrast attention module.

High-level features have rich semantic information, which can be used to guide low-level features extraction, while low-level features contain fine spatial information, which can improve the edges of high-level features. By integrating high-level and low-level features, the contrast between fore-ground and background regions can be more obvious. However, simply concatenating or uniform weighting different levels of feature maps may introduce a large amount of redundant information, which will result in blurred features and even smooth boundaries. In this paper, we propose a contrast attention (CA) to remove redundant information and highlighting edge information.

Fig. 4 presented the design of the CA module. The output feature of the $i$-th layer can be obtained as

$$D_i = cat(up(D_{i+1}), E_i^{ca})$$

for $i = 1,2,3,4$, where $cat$ denotes the channel-wise concatenation, $up$ denotes bilinear upsampling. $D_i$ represent the $i$-th decoder feature map. $E_i^{ca}$ denotes the $i$-th contrast, and it is defined as

$$E_i^{ca} = E_i - Avg(E_i)$$

where $E_i$ denotes the encoder feature map of the $i$-th layer. $Avg$ represent $Avgpool\ 3 \times 3 \times 3$.

### 2.3.2 Loss Function

The sum of dice loss and cross entropy loss are adopted as loss function. The total loss is the average of the 5 deep supervision losses.

### 2.3.3 Training and Validation Strategies

We adopted stochastic gradient descent (SGD) with Nestrov trick as the optimizer, with an initial learning rate of 0.01 and momentum 0.99. The batch size is set to be 2. We define an epoch as 250 batch iterations. The learning rate reduces in a polynomial way. The network is trained for 1000 epochs.

Five-fold cross validation is adopted. At each fold, we monitored the dice coefficients of the last 50 epochs, and selected the model with the highest dice coefficient as the final model.

### 2.3.4 Ensembling and Post-Processing

We use voting to ensemble the results of the five final models in the five-fold cross validation. Test-time augmentation is also adopted. In the first stage, we keep the largest two components, and crop the regions of interest, which is used as the input of the second stage.

# 3 Results

The network is trained on a workstation with Nvidia GTX 1080Ti GPU with 11GB memory. Due to limited memory, the batch size is set to be 2. The network is implemented on PyTorch v1.6.0. Each training epoch took about 250 second, and the training for each fold took about 84 hours. During inference, the time consumed for each subject is about 60 seconds.

Fig. 6. presented some examples of our proposed method on the test set. As we can see from (a) in Fig. 6, the overall boundary of the segmentation produced by the proposed method is relatively smooth. This is because the CAU-Net uses average pooling to avoid repeated transmission of low resolution features. Meanwhile, because the feature values of the same organization are similar, the feature values of different organizations are quite different, so the contrast attention module is also equivalent to an implicit edge attention module, which can make the model better distinguish different organizations.

Table 2. Five-fold cross validation results on the KiTS2021 training set.

| Method | # parameters | Kidney DC | Mass DC | Tumor DC | Kidney SD | Mass SD | Tumor SD |
|---|---|---|---|---|---|---|---|
| nnUNet(full) | 31.2M | 0.9666 | 0.8618 | 0.8493 | 0.9336 | 0.7532 | 0.7371 |
| nnUNet(low) | 31.2M | 0.9683 | 0.8702 | 0.8508 | 0.9272 | 0.7507 | 0.7347 |
| nnUNet(cascade) | 64.4M | **0.9747** | **0.8799** | 0.8491 | **0.9453** | **0.7714** | 0.7393 |
| Proposed | 5.6M(stage1) 18.87M(stage2) | 0.9693 | 0.8760 | **0.8509** | 0.9357 | 0.7686 | **0.7449** |

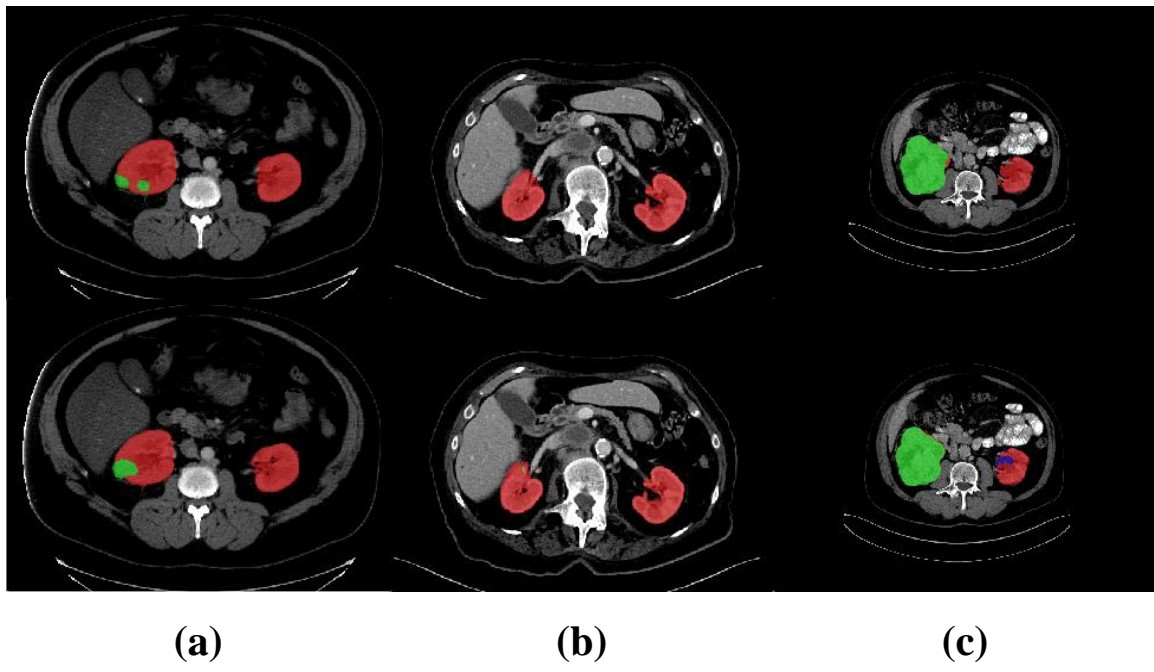

**(a)**        **(b)**        **(c)**

Fig. 6. Visualized examples of segmentation results. From top to bottom: ground truth and prediction.

The numerical evaluation results are summarized in Tab. 2. For the sake of comparison, the results of nnU-Net are also listed. As Tab.2 shows, the proposed method achieves comparable performance on most metrics, with, however, much fewer number of parameters. The proposed method has 25M parameters in total, which is only 78.3% of the nnU-Net. The proposed method achieves the highest performance in tumor segmentation, and is overall better than low-resolution nnU-Net and high-resolution nnU-Net.

## 4  Discussion and Conclusion

In this paper, a CAU-Net was proposed for KiTS2021. CA module is adopted to avoid duplication of low resolution information of features and highlighting edge information. Experimental results illustrate that the proposed method is able to achieve comparable performance with nnU-Net while using much smaller number of parameters.

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
