# OpenReview forum: "Less is more"
_MICCAI.org/2021/Challenge/KiTS — Submitted to KiTS21 Challenge_

### Official Review · Reviewer_8tA4 · 2021-08-30

**Rating:** 7

**Review:**

The authors present a coarse-to-fine approach that utilizes attention to achieve very similar performance to the nnU-Net baseline while using far fewer parameters. The paper is well-written and makes very effective use of figures to explain their relatively complex architecture. One crucial detail that is missing is the strategy that the authors used to aggregate the multiple independent instance segmentations into a composite that can be used for training and validation. This should be explicitly mentioned in the methods section.

---

### Official Review · Reviewer_jVEF · 2021-08-30

**Rating:** 7

**Review:**

### Overall

- Abstract and frontmatter look good

### Introduction

- Looks good

### Methods

- "the CNN focuses only on the ROIs and generate the segmentation map" -> "... and generate**s** the segmentation map"
- Which aggregation method did you use to sample composite masks from the individual segmentations for each instance? Most teams used majority voting. Whatever you used, you should explicitly state this in the paper
- How did you choose your bounds for clipping the HU values at -79 and 304?

### Results

- It might help the clarity of the table to bold the best metric in each column of your results table.

### Discussion and Conclusion

- Looks good

---

### Decision · Program_Chairs · 2021-08-30

**Decision:**

Minor Revisions

**Comment:**

Please address the reviewer comments and resubmit